# Preparation of Carbon Nanowall and Carbon Nanotube for Anode Material of Lithium-Ion Battery

**DOI:** 10.3390/molecules26226950

**Published:** 2021-11-17

**Authors:** Seokwon Lee, Seokhun Kwon, Kangmin Kim, Hyunil Kang, Jang Myoun Ko, Wonseok Choi

**Affiliations:** 1Department of Electrical Engineering, Hanbat National University, Daejeon 34158, Korea; dltjrdnjs000@naver.com (S.L.); kwon1567@naver.com (S.K.); talk9797@naver.com (K.K.); hikang@hanbat.ac.kr (H.K.); 2Department of Chemical and Biological Engineering, Hanbat National University, Daejeon 34158, Korea; jmko@hanbat.ac.kr

**Keywords:** carbon nanowall, carbon nanotube, PECVD, lithium-ion battery, cyclic voltammetry, FE-SEM

## Abstract

Carbon nanowall (CNW) and carbon nanotube (CNT) were prepared as anode materials of lithium-ion batteries. To fabricate a lithium-ion battery, copper (Cu) foil was cleaned using an ultrasonic cleaner in a solvent such as trichloroethylene (TCE) and used as a substrate. CNW and CNT were synthesized on Cu foil using plasma-enhanced chemical vapor deposition (PECVD) and water dispersion, respectively. CNW and CNT were used as anode materials for the lithium-ion battery, while lithium hexafluorophosphate (LiPF_6_) was used as an electrolyte to fabricate another lithium-ion battery. For the structural analysis of CNW and CNT, field emission scanning electron microscope (FE-SEM) and Raman spectroscopy analysis were performed. The Raman analysis showed that the carbon nanotube in composite material can compensate for the defects of the carbon nanowall. Cyclic voltammetry (CV) was employed for the electrochemical properties of lithium-ion batteries, fabricated by CNW and CNT, respectively. The specific capacity of CNW and CNT were calculated as 62.4 mAh/g and 49.54 mAh/g. The composite material with CNW and CNT having a specific capacity measured at 64.94 mAh/g, delivered the optimal performance.

## 1. Introduction

Lithium-ion batteries (LIB) are widely used in energy storage devices, such as portable electronic devices, which are essential in daily life. So far, graphite has been used the most as the anode material for LIB, because it has good stability during charging and discharging. When charging LIB, pulverization may occur due to a change in volume during the insertion of lithium-ions into the anode material [1]. Graphite has high stability because small volume changes occur during the charging process.

Recently, research on carbon materials and graphene as anode materials for lithium-ion batteries has been conducted [2,3,4]. There are several types of carbon allotropes [5,6], including graphene, graphite, diamond, carbon nanowalls (CNW), carbon nanotubes (CNT), and fullerenes. Graphene has a flat 2D structure in which carbon atoms form sp^2^ bonds in a hexagonal shape, while graphite is a structure in which graphene forms multiple layers by π-π bonds [7,8]. In CNW, graphene grows vertically to form a nanowall structure and carbon nanotubes are bonded by rolling graphene into a cylindrical shape. Diamond is structurally rigid because the bonds among carbon atoms are sp^3^ bonds [9].

In this paper, the characteristics of LIB, using CNW and CNT as anode materials, were investigated. CNW and CNT offer advantages of high electrical conductivity and the fast diffusion of electrons and lithium ions due to their wide surface area. The composite material of the CNW and CNT were also applied to the LIB as anode material and the characteristics were investigated.

## 2. Experimental Design

The copper (Cu) foil for the current collector needs to be cleaned because of impurities and the oxidation layer. To remove the impurities, the Cu foil was ultrasonically cleaned using TCE, acetone, methanol, and distilled water for 10 min. Figure 1a shows the schematic diagram of the Cu foil cleaning process. The oxidation layer of Cu foil, caused by exposure to atmosphere, was removed by diluted sulfuric acid.

The CNW was grown on Cu foil using plasma-enhanced chemical vapor deposition (PECVD, Woosing CryoVac, ASTEX type, 2.45 GHz microwave, Uiwang, Korea). The cleaned Cu foil was inserted in the PECVD chamber and the vacuum level of the chamber to the base pressure value 5 × 10^−5^ Torr was decreased using a vacuum pump. The mixture of H_2_ gas and CH_4_ gas was injected at a ratio of 2:1 into the chamber with the Cu foil heated to 700 °C. The injected gases were ionized by plasma power of 1300 W and the carbon atoms were deposited on Cu foil to synthesize CNW. Figure 2 shows the schematic diagram of the PECVD with the process of growing CNW. In addition, Table 1 shows the conditions for growing the CNW via PECVD.

Using the MWCNTs water dispersion (3 wt%, >95+%, OD: 5–15 nm, Length 50 μm), purchased from US Research Nanomaterials, the CNT was synthesized on Cu foil. A MWCNT water dispersion solution of 3 mL was dropped onto the Cu foil by pipette and dried by hotplate at 70 °C for 20 min. Figure 1b shows the schematic diagram of synthesizing CNT using the water dispersion method.

The composite material of CNW and CNT was prepared in two steps. The CNW was grown on Cu foil by PECVD, and the CNT was synthesized on CNW using the water dispersion method. Figure 1c shows the schematic diagram fot synthesizing the composite material with CNW and CNT.

To investigate the surface and cross-section structure, the Raman shift (micro-Raman Spectrometer, FEX, Seongnam, Korea) and field emission scanning electron microscopy (FE-SEM, Hitachi S-4800) image were analyzed for each sample. In the Raman analysis, the excitation laser wavelength was ~531 nm, the excitation laser power was ~0.3 mW and the spectral resolution was ~1.9 to 2.1/cm. To examine the electrochemical properties of anode materials prepared by CNW and CNT, the coin cell (CR2032, Wlcos, Gunpo, Korea) was manufactured. The lithium metal (150 μm thickness) was used as a counter electrode, while the CNW and CNT were used as working electrodes. All the coin cells used 1 M LiPF_6_ (Panax Etec) in EC/DMC = 4/6 (*v*/*v*) as an electrolyte, while polyethylene (Celgard, 20 μm) was used as the separator. Cyclic voltammetry (CV, WizECM-1200, Daejeon, Korea) was performed to measure the specific capacity of the anode materials prepared by CNW and CNT.

## 3. Results and Discussion

Figure 3 shows the FE-SEM surface and cross-section images of each sample. Figure 3a is the FE-SEM surface image of the CNW, and it can be confirmed that the wall structure of the CNW was well grown by PECVD. The cylindrical structure of CNT was also well synthesized on Cu foil using the water dispersion method. This can be confirmed in the FE-SEM surface image in Figure 3b. Figure 3c–e show the FE-SEM cross-section image of CNW, CNT, and composite material, respectively. It can be confirmed that CNT was well synthesized on the CNW as composite material.

Raman analysis was performed to investigate the structure of CNW, CNT and composite material with CNW and CNT. In Figure 4a, the Raman shift for each sample shows the D peak, G peak, D’ peak and 2D peak. A D peak indicates when a defect in the sample was observed at 1349 cm^−1^. The G peak, which is often observed in graphene, such as those found in carbon materials, was observed at 1582 cm^−1,^ while 2D peak was observed at 2697 cm^−1^ as a peak affected by the π-π bond in the graphene [10]. In contrast to CNTs, it can be seen that the D peak of CNWs were measured with a particularly strong intensity, which can be attributed to the edge of graphene with vertically grown wall-shaped structures in CNW [11,12]. A strong D peak was observed in the Raman shift of CNW, but a D peak with similar intensity to that of CNT was observed in the composite material. It seems that the wall-shaped structure of the CNW, which was the cause of the D peak, was reduced due to the bonding with the CNT in the composite material. The I_D_/I_G_ and I_2D_/I_G_ ratio of CNW, CNT, and the composite material, calculated based on the results of Raman shift intensity, are shown in Figure 4b. The I_D_/I_G_ ratio, which indicates the defects of the sample, also shows a high value due to the wall-shaped structure of the CNW, while the I_2D_/I_G_ ratio represents the thickness of graphene [13]. The lower the I_2D_/I_G_ ratio is, the thicker graphite is grown by PECVD. By the I_2D_/I_G_ ratio of each sample in Figure 4b, all of the CNW and CNT consisted of multi-layered graphene.

Cyclic voltammetry (CV) was performed with a typical coin cell and working electrode made from CNW, CNT, and composite material, respectively. The CV was measured in a 0~1.7 V potential window and the scan rate was 0.1 mV/s for 12th cycle. Figure 5 shows the CV graph of CNW, CNT, and composite material for each cycle. In the CV graph, the oxidation peak indicates the charging process of the LIB since lithiation occurs in anode materials. The CV graph of CNW is shown in Figure 5a, representing one oxidation peak at 0.11 V. In Figure 5b, the CV graph of the CNT shows the three oxidation peaks at 0.25 V, 0.42 V, and 0.54 V. Figure 5c shows the CV graph of the composite material with CNW and CNT, and there are two oxidation peaks at 0.18 V and 0.53 V. The reduction peak in the CV graph indicates the discharging process of the LIB since delithiation occurs in anode materials.

Figure 5a shows the two reduction peaks at 0.03 V and 0.53 V. Figure 5b shows one reduction peak at 0.51 V, while Figure 5c shows the one reduction peak at 0.51 V. Although there is a slight increase, Figure 5a shows the increment of the oxidation peak according to the cycle number in the CV graph of CNW. In Figure 5b, the CV graph of the CNT also shows the increment of the oxidation peaks according to cycle number. However, in Figure 5c, the CV graph of composite material with CNW and CNT does not show the increment of the oxidation peaks. In the CV graph, the increment of the oxidation peak indicates the increment of the specific capacity and good stability in the charging process. In the paper reported by Lin et.al, a CV graph with a similar trend was presented for graphene nanowalls [14].

The specific capacity of the LIB can be calculated in the CV graph by the internal area, mass of the anode materials, and scan rate. The internal area of the CV graph can be confirmed using the Origin program. *C_p_* [mAh/g] is the specific capacity of the LIB, *A* [VA] is the internal area of the CV graph, *m* is the mass of the anode materials, and *k* [mV/s] is the scan rate of the CV graph. The equation for calculating the specific capacity of the LIB is shown below [15].
(1)$$C_{p}=\frac{1}{2\;km}\int i\left(v\right)dv\;\left[As/\mathrm{mg}\right]=\frac{1000\;A}{7.2\;mk}\;\left[\mathrm{mAh}/g\right]$$

The specific capacities for each sample, calculated using Equation (1), are shown in Figure 6. In the CV graphs of the CNW and CNT, the increment of the peaks are shown according to the cycle number. This way, the tendency of each specific capacity is to increase according to the cycle number. In contrast, the composite material, with CNW and CNT of a specific capacity, shows a declining tendency by the slope −0.097. The average specific capacity in 12 cycles is 62.4 mAh/g for CNW, 49.54 mAh/g for CNT, and 64.94 mAh/g for the composite material. The internal area of CNW is very small compared to that of CNT in Figure 5d, this is because the mass of the CNW is also very small. Small mass is at an advantage, and it indicates that more lithium-ions can be stored in the small mass of CNW. The specific capacity of the composite material was calculated to be higher than that of CNW and CNT. As such, using CNW and CNT as a composite material has the advantage of high specific capacity. The composite material can store more lithium-ions by compensating CNT for large defects in the wall structure of CNW, resulting in the increment of specific capacity.

## 4. Conclusions

In this work, the CNW, CNT, and composite material are prepared as anode materials to investigate the properties of LIB. FE-SEM image and Raman analysis were employed to examine the structure of each sample. In Raman shift, we found that the CNT synthesized on CNW can reduce defects that occur in the wall structure of CNW. Coin cells were manufactured by each sample to investigate the electrochemical properties. CV analysis was performed on the coin cells to calculate the specific capacity of each cell. Once the specific capacity calculated using the CV graph for each cycle, we noticed that the composite material with CNW and CNT is represents the best specific capacity. In the composite material, the CNT can compensate for the defect of CNW that has the advantage of storing many lithium-ions in a small mass. For this reason, the composite material’s specific capacity was shown to be optimal.

## Figures and Tables

**Figure 1 molecules-26-06950-f001:**
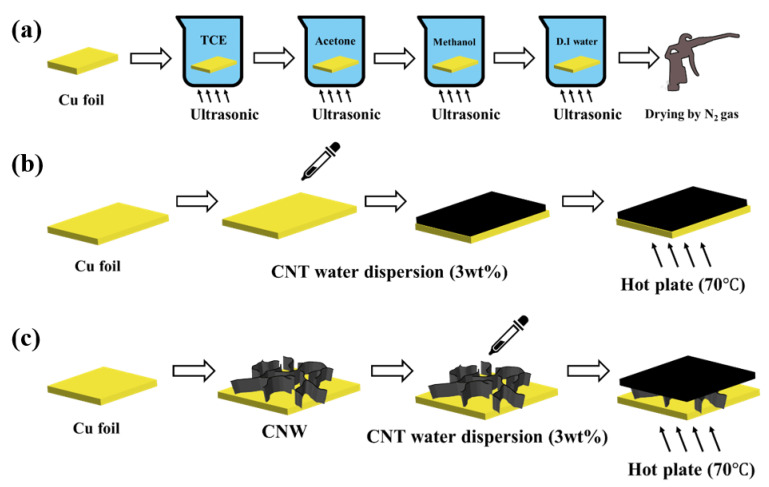
(**a**) The schematic diagram of the Cu foil cleaning process, (**b**) synthesizing the CNT using water dispersion method and (**c**) synthesizing the composite material with CNW and CNT.

**Figure 2 molecules-26-06950-f002:**
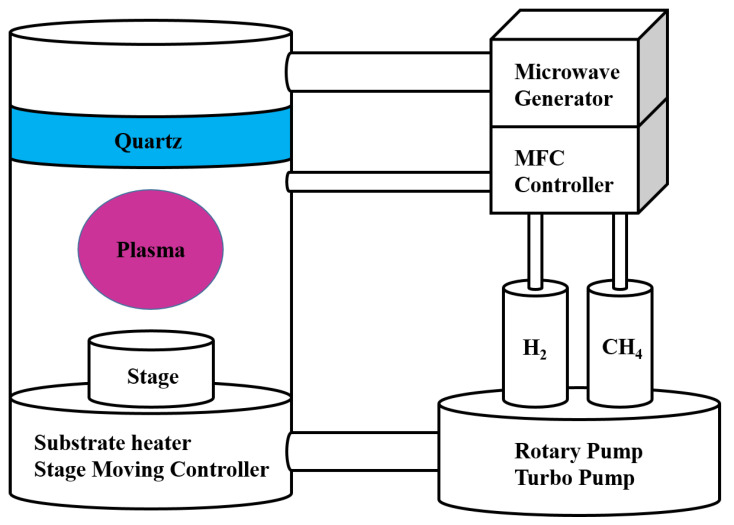
The schematic diagram of PECVD.

**Figure 3 molecules-26-06950-f003:**
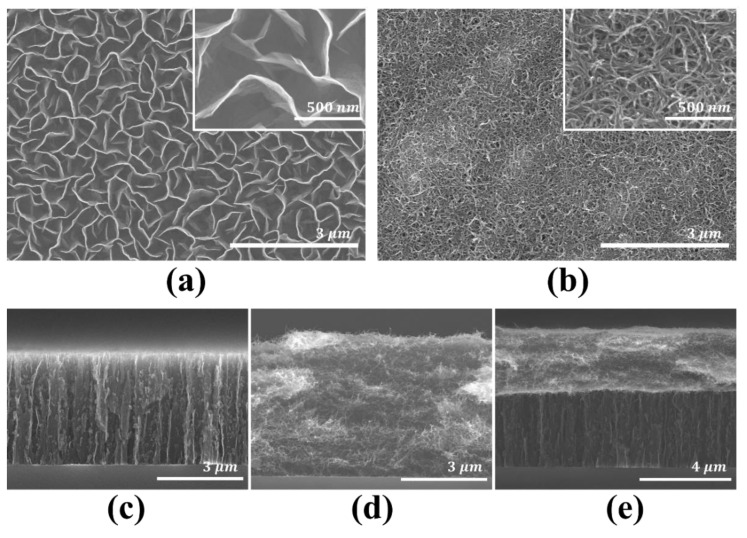
The FE-SEM surface image of the (**a**) CNW (scale bar, 3 μm) with enlarged CNW (inset, scale bar, 500 nm) and (**b**) CNT (scale bar, 3 μm) with enlarged CNT (inset, scale bar, 500 nm). The FE-SEM cross-section image of the (**c**) CNW (scale bar, 3 μm), (**d**) CNT (scale bar, 3 μm) and (**e**) composite material with CNW and CNT (scale bar, 4 μm).

**Figure 4 molecules-26-06950-f004:**
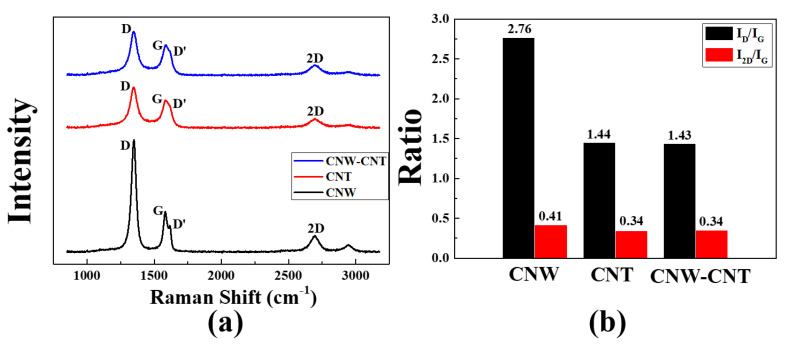
(**a**) The Rama shift of the CNW (black line), CNT (red line), and composite material with CNW and CNT (blue line). (**b**) The I_D_/I_G_ and I_2D_/I_G_ ratio of CNW, CNT and composite material from Raman shift.

**Figure 5 molecules-26-06950-f005:**
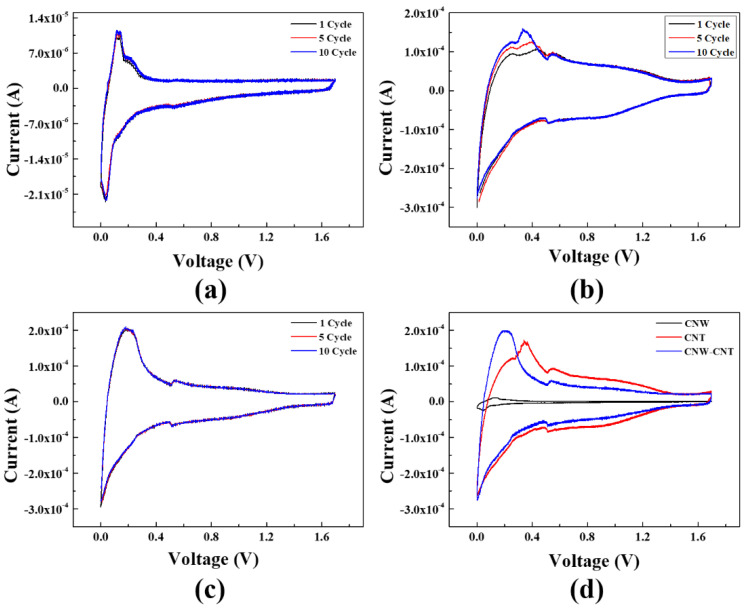
The CV graphs of (**a**) CNW, (**b**) CNT and (**c**) composite material for 1, 5, 10 cycle. (**d**) The CV graph of the 12th cycle for each of the samples.

**Figure 6 molecules-26-06950-f006:**
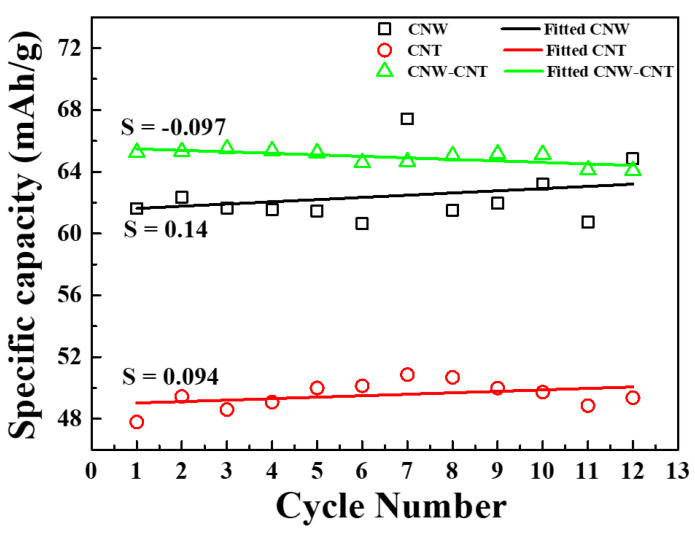
The Specific capacity of the CNW, CNT, and composite material for 12 cycles.

**Table 1 molecules-26-06950-t001:** The conditions of growing CNW via PECVD.

Substrate	Base Pressure	Working Pressure	Chamber Atmosphere	Temperature	Growth Time
Cu foil	5 × 10^−5^ Torr	4 × 10^−2^ Torr	H_2_:CH_4_ = 2:1	700 °C	15 min

## Data Availability

The data presented in this study are available from the corresponding author on reasonable request.

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
