# Peer review of "Preparation of Carbon Nanowall and Carbon Nanotube for Anode Material of Lithium-Ion Battery"

_molecules, 2021, doi:10.3390/molecules26226950_

Round 1

Reviewer 1 Report

Choi et al prepared carbon nanowall and carbon nanotube for anode material of lithium-ion battery. The experimental results and related findings have important reference significance for the preparation of other carbon materials. The work is interesting and has a strong appeal to the readers. The paper can be considered for acceptance after a minor revision.

1. Carbon materials with different structures have been prepared by various methods. The background of these carbon materials as well as their preparation routes should be detailedly stated in the introduction section. In addition, some references are too old, it is suggested to cite more literatures in the past two years. The following recent research work can be used for reference.

[1] Hard Carbon Anodes for Next-Generation Li-Ion Batteries: Review and Perspective, Advanced Energy Materials, 2021, 2101650.  doi: 10.1002/aenm.202101650
[2] Hierarchically Porous Carbon Derived from Biomass Reed Flowers as Highly Stable Li-Ion Battery Anode, Nanomaterials 2020, 10, 346. doi:10.3390/nano10020346
[3] Superior carbon black: High-performance anode and conducting additive for rechargeable Li- and Na-ion batteries, Chemical Engineering Journal, 2021, 417, 129242.  doi:10.1016/j.cej.2021.129242

2. More detailed characterization of the material, including XRD and TEM, is suggested to provide in the revision.

3. The authors calculated the specific capacity of the LIB through the CV graph by the internal area, mass of the anode materials and scan rate. Galvanostatic charge-discharge curves, cyclic and rate performance of the materials are suggested to test through battery tester for commonly comparation with other studies.

4. The advantages of the as-synthesized material over other materials need to be emphasized.

Author Response

  • Answering letter. …………………………………………………………

Title: Preparation of carbon nanowall and carbon nanotube for anode material of lithium-ion battery

Authors: Seokwon Lee, Seokhun Kwon, Kangmin Kim, Hyunil Kang, Jang Myoun Ko and Wonseok Choi

Reviewer #1:

Choi et al prepared carbon nanowall and carbon nanotube for anode material of lithium-ion battery. The experimental results and related findings have important reference significance for the preparation of other carbon materials. The work is interesting and has a strong appeal to the readers. The paper can be considered for acceptance after a minor revision.

  1. Carbon materials with different structures have been prepared by various methods. The background of these carbon materials as well as their preparation routes should be detailedly stated in the introduction section. In addition, some references are too old, it is suggested to cite more literatures in the past two years. The following recent research work can be used for reference.

[1] Hard Carbon Anodes for Next-Generation Li-Ion Batteries: Review and Perspective, Advanced Energy Materials, 2021, 2101650.  doi: 10.1002/aenm.202101650

[2] Hierarchically Porous Carbon Derived from Biomass Reed Flowers as Highly Stable Li-Ion Battery Anode, Nanomaterials 2020, 10, 346. doi:10.3390/nano10020346

[3] Superior carbon black: High-performance anode and conducting additive for rechargeable Li- and Na-ion batteries, Chemical Engineering Journal, 2021, 417, 129242.  doi:10.1016/j.cej.2021.129242

→ The mentioned references are add to reference number 2-4.

  1. More detailed characterization of the material, including XRD and TEM, is suggested to provide in the revision.

→ Currently, our laboratory does not have the equipment and environment for XRD and TEM analysis, and there is not enough time to request analysis. In addition, since TEM analysis of carbon nanowalls is very difficult, it is estimated that it will be difficult to add TEM analysis contents to the revisions.

  1. The authors calculated the specific capacity of the LIB through the CV graph by the internal area, mass of the anode materials and scan rate. Galvanostatic charge-discharge curves, cyclic and rate performance of the materials are suggested to test through battery tester for commonly comparation with other studies.

→ We already know that the galvanostatic charge-discharge test is being used to evaluate battery characteristics, unfortunately, it seems difficult to analyze the characteristics of LIB because it takes very long elapsed time. In the next carbon nanowall-related LIB research, the galvanostatic charge-discharge test will be conducted.

  1. The advantages of the as-synthesized material over other materials need to be emphasized.

→ In the description of Figure 6, we add sentence emphasized advantages for the as-synthesized materials of “There is an advantage that the specific capacity is higher to apply composite material than to apply CNW and CNT respectively”.

We appreciate to your suggestions and questions.

Reviewer 2 Report

The manuscript is dealing with the use of carbon based nanostructured materials in anodes of lithium -ion batteries (LIB).

In general the topic is interesting and timely as LIBs are at great attention, and fits to the interest of the Journal and the Special Issue.

More specifically Authors present a comparative analysis of use of carbon nanowall (CNW), carbon nanotubes (CNT), and also a composite form of CNW-CNT as anode materials in LIB. The paper is focused entirely on the fabrication aspects (preparation as indicated in the title) of CNT, CNW, and composite CNW-CNT, as well as on their characterization and analysis by means of FE-SEM, Raman Spectrospcopy, and finally cyclic voltametry for electrochemical characterization. Those experimental results are well organized and presented, while the conclusions are supported by the findings.

However I believe that the manuscript in order to be more appealing  it needs some more and considerable work in terms of critical assessment of results, comparison with prior results in literature, and a critical discussion on the potential impact and applications.

Therefore I would propose the following

1) Update and include more references on the use on CNTs, CNW, graphene as anode materials in LIB. The listed references are very limited and inadequate.

2) State clearly the novelty presented at the manuscript. Is the CNW-CNT composite presented as the first demonstration?

3) Describe the reason CNW-CNT composite was proposed.  Describe clearly or provide intuitive reason for this proposal.

4) What is finally the advantage of CNW-CNT combined action?  This is refereed only very briefly while authors need to expand the discussion and relate to the presented results.

5)  Is the performance better that other carbon based materials, like graphene ? Is the performance adequate enough to compensate the additional cost of two step process? Try to quantify if possible.

6) The finding and the assessment are not presented clearly at the Abstract section. Please rewrite the abstract by including the results in a critical way and not only stating the preparation process.

Author Response

  • Answering letter. …………………………………………………………

Title: Preparation of carbon nanowall and carbon nanotube for anode material of lithium-ion battery

Authors: Seokwon Lee, Seokhun Kwon, Kangmin Kim, Hyunil Kang, Jang Myoun Ko and Wonseok Choi

Reviewer #2:

The manuscript is dealing with the use of carbon based nanostructured materials in anodes of lithium -ion batteries (LIB).

In general the topic is interesting and timely as LIBs are at great attention, and fits to the interest of the Journal and the Special Issue.

More specifically Authors present a comparative analysis of use of carbon nanowall (CNW), carbon nanotubes (CNT), and also a composite form of CNW-CNT as anode materials in LIB. The paper is focused entirely on the fabrication aspects (preparation as indicated in the title) of CNT, CNW, and composite CNW-CNT, as well as on their characterization and analysis by means of FE-SEM, Raman Spectrospcopy, and finally cyclic voltametry for electrochemical characterization. Those experimental results are well organized and presented, while the conclusions are supported by the findings.

However I believe that the manuscript in order to be more appealing  it needs some more and considerable work in terms of critical assessment of results, comparison with prior results in literature, and a critical discussion on the potential impact and applications.

Therefore I would propose the following

1) Update and include more references on the use on CNTs, CNW, graphene as anode materials in LIB. The listed references are very limited and inadequate.

→ The references about carbon materials are added to reference number 2-4.

2) State clearly the novelty presented at the manuscript. Is the CNW-CNT composite presented as the first demonstration?

→ No other papers didn’t performed that CNW-CNT composite material is fabricated to lithium-ion battery. We think that there is novelty because the CNW-CNT composite material is first presentation in our manuscript.

3) Describe the reason CNW-CNT composite was proposed.  Describe clearly or provide intuitive reason for this proposal.

→ The purpose of experiment for composite material with CNW and CNT is to get the both advantages of CNW and CNT.

4) What is finally the advantage of CNW-CNT combined action?  This is refereed only very briefly while authors need to expand the discussion and relate to the presented results.

→ In the description of Figure 6, we add sentence emphasized advantages for the as-synthesized materials of “There is an advantage that the specific capacity is higher to apply composite material than to apply CNW and CNT respectively”.

5)  Is the performance better that other carbon based materials, like graphene ? Is the performance adequate enough to compensate the additional cost of two step process? Try to quantify if possible.

→ The performance of composite material with CNW and CNT is worse than graphene those specific capacity is 372 mAh/g. However, carbon nanowalls and carbon nanotubes can be directly deposited on copper foil and other current collectors, and unlike graphene, which is destroyed at high temperatures, carbon nanowalls and carbon nanotubes have heat resistance, so it is judged that they are useful in extreme situations.

6) The finding and the assessment are not presented clearly at the Abstract section. Please rewrite the abstract by including the results in a critical way and not only stating the preparation process.

→ In the abstract, the two sentences representing the results of “From the Raman analysis, it can be seen that carbon nanotube in composite material can compensate for the defects of carbon nanowall” and “The composite material with CNW and CNT of specific capacity measured as 64.94 mAh/g represented the best performance” are added.

We appreciate to your suggestions and questions.

Reviewer 3 Report

  1. What is new in your contribution? The novelty should be underlined.
  2. The specific capacity is rather low and below the current state of the art.
  3. Results/conclusions are not supported by relevant references.
  4. In the experimental section the full protocol should be described (e.g., for Raman analysis laser wavelength, detectors; for SEM sample preparation, electron beam....). The resolution /magnification is not presented clearly.
  5. ID/IG ratio is calculated after deconvolution treatment?...be precise
  6. Figure 7 (d), this is because the mass of the CNW is also very small ...there are only 6 images in the manuscript, please refer to the proper one.
  7. You should make multiple cycles, 10 is not enough....
  8. There are several typos....please revise the whole manuscript
  9. Galvanostatic discharge/charge voltage profiles and EIS measurements are missing

Author Response

  • Answering letter. …………………………………………………………

Title: Preparation of carbon nanowall and carbon nanotube for anode material of lithium-ion battery

Authors: Seokwon Lee, Seokhun Kwon, Kangmin Kim, Hyunil Kang, Jang Myoun Ko and Wonseok Choi

Reviewer #3:

1.What is new in your contribution? The novelty should be underlined.

→ No other papers didn’t performed that CNW-CNT composite material is fabricated to lithium-ion battery. We think that there is novelty because the CNW-CNT composite material is first presentation in our manuscript. We can underline the sentences in the introduction of “The composite material of the CNW and CNT were also applied to the LIB as anode material and the characteristics were investigated”.

2.The specific capacity is rather low and below the current state of the art.

→ The performance of composite material with CNW and CNT is worse than graphene those specific capacity is 372 mAh/g. However, carbon nanowalls and carbon nanotubes can be directly deposited on copper foil and other current collectors, and unlike graphene, which is destroyed at high temperatures, carbon nanowalls and carbon nanotubes have heat resistance, so it is judged that they are useful in extreme situations.

3.Results/conclusions are not supported by relevant references.

→ The additional reference is added to results and discussion part. And also the sentence is added in “In the paper reported by Lin et.al, CV graph with a similar trend was measured for graphene nanowalls [13]”.

4.In the experimental section the full protocol should be described (e.g., for Raman analysis laser wavelength, detectors; for SEM sample preparation, electron beam....). The resolution /magnification is not presented clearly.

→ In experimental design, the details of Raman analysis are added with sentence of “In Raman analysis, the excitation laser wavelength was ~531 nm, the excitation laser power was ~0.3 mW and the spectral resolution was ~1.9 to 2.1 /cm”. In figure 3, The scale bar that was not clearly marked on the picture was revised, and the scale bar information was added to the picture description field.

5.ID/IG ratio is calculated after deconvolution treatment?...be precise

→ When calculating the ID/IG ratio, it was calculated based on the Raman shift intensity without the deconvolution treatment. In results and discussion, the sentence is revised to “The ID/IG and I2D/IG ratio of CNW, CNT and composite material calculated based on the results of Raman shift intensity are shown in Figure 4 (b)”.

6.Figure 7 (d), this is because the mass of the CNW is also very small ...there are only 6 images in the manuscript, please refer to the proper one.

→ In results and discussion, the sentence is added in “Small mass is the one of the advantage and it is indicating the more lithium-ions can be storage in small mass of CNW”. The figure 7 (d) is the typo and we correct the figure 7 (d) to figure 5 (d).

7.You should make multiple cycles, 10 is not enough....

→ In general CV analysis or galvanostatic charge-discharge test, 50 or 100 cycles of analysis are in progress, but in our manuscript, scan rate of 0.1 mV/sec is adopted for more detailed CV characteristic analysis, so it is difficult to analysis more cycles for LIBs because of the very long elapsed time.

8.There are several typos....please revise the whole manuscript

→ After reading the entire manuscript thoroughly, all typos found were corrected.

9.Galvanostatic discharge/charge voltage profiles and EIS measurements are missing

→ We already know that the galvanostatic charge-discharge test is being used to evaluate battery characteristics, unfortunately, it seems difficult to analyze the characteristics of LIB because it takes very long elapsed time. In the next carbon nanowall-related LIB research, the galvanostatic charge-discharge test will be conducted.

We appreciate to your suggestions and questions.

Round 2

Reviewer 2 Report

Authors have clarified some of the raised issues in their reply letter, however not at a required and expected level of analysis and elaboration. Most importantly, they have not included those clarifications and additional discussion in the actual manuscript in order to improve the presentation of the paper. The revised manuscript is only marginally improved.

I suggest them to do so. I suggest also to provide a discussion on the CNW and CNT role and their combination as this is the main claimed novelty of the paper How are CNW and CNT functioning and what their combination provides as an advantage. The capacity pf the composite CNW/CNT is only marginally higher than that of CNW and a discussion of cost /benefit needs to be provided.

Finally, Authors needs to drastically improve the quality of English especially at the newly added material.

Author Response

  • Answering letter. ……………………………………………………………………………

Title: Preparation of carbon nanowall and carbon nanotube for anode material of lithium-ion battery

Authors: Seokwon Lee, Seokhun Kwon, Kangmin Kim, Hyunil Kang, Jang Myoun Ko and Wonseok Choi

Reviewer #2:

Authors have clarified some of the raised issues in their reply letter, however not at a required and expected level of analysis and elaboration. Most importantly, they have not included those clarifications and additional discussion in the actual manuscript in order to improve the presentation of the paper. The revised manuscript is only marginally improved.

I suggest them to do so. I suggest also to provide a discussion on the CNW and CNT role and their combination as this is the main claimed novelty of the paper How are CNW and CNT functioning and what their combination provides as an advantage. The capacity pf the composite CNW/CNT is only marginally higher than that of CNW and a discussion of cost /benefit needs to be provided.

Finally, Authors needs to drastically improve the quality of English especially at the newly added material.

→ We think that the purpose and significance of our manuscript is attempt of use carbon nanowalls and carbon nanotubes together as anode materials in LIB. One of the advantages of using carbon nanowalls and carbon nanotubes together is that carbon nanotubes can compensate for the D peak of Raman shift caused by the wall structure of carbon nanowall. At the same time, the specific capacity of the composite material can be increased by taking advantage of the lower weight of carbon nanowall can store more lithium ions. In the part of conclusion, the advantages of the composite material with CNW and CNT are revised in sentence of “ In the composite material, the CNT can compensate for the defect of the CNW that has the advantage of storing many lithium-ions in a small mass, and this why the specific capac-ity of composite material is shown to be optimal”. A request for English proofreading of the revised and added sentences will be carried out, and upon completion, a proof of proofreading will be attached.

We appreciate to your suggestions and questions.

Reviewer 3 Report

1.What is new in your contribution? The novelty should be underlined.

→ No other papers didn’t performed that CNW-CNT composite material is fabricated to lithium-ion battery. We think that there is novelty because the CNW-CNT composite material is first presentation in our manuscript. We can underline the sentences in the introduction of “The composite material of the CNW and CNT were also applied to the LIB as anode material and the characteristics were investigated”.

Response: Going this way we can say that each material is a new one… You can also mix together CNT and porous carbon and apply it for batteries…

3.Results/conclusions are not supported by relevant references.

→ The additional reference is added to results and discussion part. And also the sentence is added in “In the paper reported by Lin et.al, CV graph with a similar trend was measured for graphene nanowalls [13]”.

Response: This is a general remark, 3 more references are not enough...

4.In the experimental section the full protocol should be described (e.g., for Raman analysis laser wavelength, detectors; for SEM sample preparation, electron beam....). The resolution /magnification is not presented clearly.

→ In experimental design, the details of Raman analysis are added with sentence of “In Raman analysis, the excitation laser wavelength was ~531 nm, the excitation laser power was ~0.3 mW and the spectral resolution was ~1.9 to 2.1 /cm”. In figure 3, The scale bar that was not clearly marked on the picture was revised, and the scale bar information was added to the picture description field.

Response: This is a general remark, e.g., electrochemical setup should be further described... I also agree that XRD spectra can support the Raman measurements!

7.You should make multiple cycles, 10 is not enough....

→ In general CV analysis or galvanostatic charge-discharge test, 50 or 100 cycles of analysis are in progress, but in our manuscript, scan rate of 0.1 mV/sec is adopted for more detailed CV characteristic analysis, so it is difficult to analysis more cycles for LIBs because of the very long elapsed time.

 Response: you can choose either conduct research partially or provide the full characterization even if "it seems difficult to analyze the characteristics of LIB because it takes very long elapsed time". It will increase the interest to the readers.

9.Galvanostatic discharge/charge voltage profiles and EIS measurements are missing

→ We already know that the galvanostatic charge-discharge test is being used to evaluate battery characteristics, unfortunately, it seems difficult to analyze the characteristics of LIB because it takes very long elapsed time. In the next carbon nanowall-related LIB research, the galvanostatic charge-discharge test will be conducted.

response: Please see above.

Author Response

  • Answering letter. ……………………………………………………………………………

Title: Preparation of carbon nanowall and carbon nanotube for anode material of lithium-ion battery

Authors: Seokwon Lee, Seokhun Kwon, Kangmin Kim, Hyunil Kang, Jang Myoun Ko and Wonseok Choi

Reviewer #3:

  1. What is new in your contribution? The novelty should be underlined.

→ No other papers didn’t performed that CNW-CNT composite material is fabricated to lithium-ion battery. We think that there is novelty because the CNW-CNT composite material is first presentation in our manuscript. We can underline the sentences in the introduction of “The composite material of the CNW and CNT were also applied to the LIB as anode material and the characteristics were investigated”.

Response: Going this way we can say that each material is a new one… You can also mix together CNT and porous carbon and apply it for batteries…

→ A single carbon material has been used in the research of LIBs so far, but in our manuscript, it is significant that two carbon materials are used together.

  1. Results/conclusions are not supported by relevant references.

→ The additional reference is added to results and discussion part. And also the sentence is added in “In the paper reported by Lin et.al, CV graph with a similar trend was measured for graphene nanowalls [14]”.

Response: This is a general remark, 3 more references are not enough...

→ The reference was added in number 12, in order to support that the wall structure of carbon nanowall affects the D peak in the Raman shift.

  1. In the experimental section the full protocol should be described (e.g., for Raman analysis laser wavelength, detectors; for SEM sample preparation, electron beam....). The resolution /magnification is not presented clearly.

→ In experimental design, the details of Raman analysis are added with sentence of “In Raman analysis, the excitation laser wavelength was ~531 nm, the excitation laser power was ~0.3 mW and the spectral resolution was ~1.9 to 2.1 /cm”. In figure 3, The scale bar that was not clearly marked on the picture was revised, and the scale bar information was added to the picture description field.

Response: This is a general remark, e.g., electrochemical setup should be further described... I also agree that XRD spectra can support the Raman measurements!

→ The electrochemical setup also added to experimental design with the sentence of “The lithium metal (150 μm thickness) was used as counter electrode while the CNW and CNT were used as working electrode. All the coin cells used 1M LiPF6 (Panax Etec) in EC/DMC = 4/6 (v/v) as electrolyte and the polyethylene (Celgard, 20 μm) was used as separator”.

  1. You should make multiple cycles, 10 is not enough....

→ In general CV analysis or galvanostatic charge-discharge test, 50 or 100 cycles of analysis are in progress, but in our manuscript, scan rate of 0.1 mV/sec is adopted for more detailed CV characteristic analysis, so it is difficult to analysis more cycles for LIBs because of the very long elapsed time.

 Response: you can choose either conduct research partially or provide the full characterization even if "it seems difficult to analyze the characteristics of LIB because it takes very long elapsed time". It will increase the interest to the readers.

→ We appreciate your advice and comments and understand the importance of cyclic voltammetry and galvanostatic charge-discharge test. Unfortunately, however, we have only been given 5 days to make major revision, and the analysis can not be performed anymore since the LIB samples are disposed. Therefore, when the next study proceeds, we will strongly accept your advice and attach the long cycle of cyclic voltammetry and galvanostatic charge-discharge test.

  1. Galvanostatic discharge/charge voltage profiles and EIS measurements are missing

→ We already know that the galvanostatic charge-discharge test is being used to evaluate battery characteristics, unfortunately, it seems difficult to analyze the characteristics of LIB because it takes very long elapsed time. In the next carbon nanowall-related LIB research, the galvanostatic charge-discharge test will be conducted.

Response: Please see above.

We appreciate to your suggestions and questions.